# Growth Performance, Histological Changes and Functional Tests of Broiler Chickens Fed Diets Supplemented with *Tribulus Terrestris* Powder

**DOI:** 10.3390/ani12151930

**Published:** 2022-07-28

**Authors:** Maged A. Al-Garadi, Hani H. Al-Baadani, Abdulmohsen H. Alqhtani

**Affiliations:** Department of Animal Production, College of Food and Agriculture Science, King Saud University, P.O. Box 2460, Riyadh 11451, Saudi Arabia; malgaradi@ksu.edu.sa (M.A.A.-G.); hanee7811@gmail.com (H.H.A.-B.)

**Keywords:** *Gallus domesticus*, *Tribulus Terrestris*, histology, liver, kidney, intestine

## Abstract

**Simple Summary:**

*Tribulus Terrestris* (TT) is extremely rich in substances with potential biological activity and is therefore considered to be a promising ingredient in industrial medicinal preparations based on its saponin fraction. On the other hand, studies regarding broiler chickens are limited. However, the aim of this study was to verify the safe amount of TT powder that can be used as a medicinal plant in the diet of broiler chickens by determining the growth performance and histological changes and functional tests of some target organs. In conclusion, 0.75 g/kg TT is the best dosage for the health of broiler chickens. There is a need for low dose studies to identify the mechanism of the pharmacological effect of TT on broilers.

**Abstract:**

The current experiment aimed to investigate the effects of TT powder on performance parameters and functional tests, as well as on morphological and histological changes in the liver, kidney and ileum in broiler chickens. Commercial broilers (total = 168 females) were used, equally divided into three dietary treatments (C = 0.0, T1 = 0.75, and T2 = 1.5 g/kg diet). The growth performance (1–35 days of age), absolute and relative weight, liver and kidney functional tests, intestinal morphology (14 and 35 days of age), and histomorphology of the ileum (35 days of age) were evaluated. At 35 days of age, histopathological changes in the ileum, liver, and kidney were also examined. The results showed that the growth performance and absolute and relative weights of the liver and kidney had no negative effects when dietary supplementation with TT powder was given at 0.75 g/kg diet (T1), whereas a decrease was observed at T2 (*p* < 0.05). Liver and kidney functional tests showed no significant effects in all feed treatments (14 days), while T1 showed lower (*p* < 0.05) ALT and AST levels (35 days). T1 exhibited higher weights, lengths, and weight-to-length ratios of the small intestine, and relative lengths of the duodenum (*p* < 0.05). Histomorphometric measurements of the ileum were higher (*p* < 0.05) in chickens fed the 0.75 g TT/kg diet, and except for in the goblet cell count and epithelial thickness, there were no differences between treatments (*p* > 0.05). In T1, hepatocytes were more normal but hepatic sinusoids were dilated, whereas in T2, lymphocytes had infiltrated around the central vein and lining endothelial cells had been lost. The kidney was improved in T1 and T2 compared with the control group. Ileal villi were shorter in T2, and some villi fused with enterocyte necrosis and inflammatory cells accumulated in the lumen. We concluded that TT powder (0.75 g/kg feed) has a safe effect and is healthy for broilers.

## 1. Introduction

Medicinal herbal and aromatic plants have been a gaining interest in recent years as feed additives in poultry. Medicinal herbs can alter the active compounds in final products depending on the part used (e.g., seed, leaf, root, or bark) and the processing technique (e.g., extraction with non-aqueous solvents) [1,2]. Several studies in recent years have shown that the use of medicinal herbs as feed additives can improve the health, performance, and nutrient digestibility of broiler chickens [3,4,5]. 

*Tribulus Terrestris* (TT) belongs to the Zygophyllaceae family [6]. It is widely distributed in many areas (tropical and temperate regions of the world), especially in northeastern Yemen and southern Saudi Arabia, and it competes with most crops in the early stage of development [7]. TT is a medicinal plant accepted and recognized by humans as a traditional remedy for many diseases in the form of the whole plant or especially the fruits [4]. Rathod et al. [8]; Daniyal et al. [9] reported that the phytotherapeutic potential of TT is present in the leaves, roots, seeds, and fruits. TT is extremely rich in substances with potential biological activity such as steroid glycosides (saponins), protodioscin, glycosides, alkaloids and flavonoids [10]. Additionally, Khalid et al. [1] showed that the pharmacologically active components of TT belong to flavonoids and phenols. However, it is considered as an ingredient in the industrial production of medicinal preparations based on its saponin fraction [11]. Therefore, TT and its extracts are used as medicinal plants to enhance performance and digestibility, to remove kidney stones, as a diuretic, to treat urinary tract infections, hypercholesterolemia, and diabetes, and to improve liver health [12,13,14,15]. In addition, the fruit of TT powder is widely and effectively used as a feed additive to improve the reproductive and health potential of roosters [16], mice [17], and humans [18]. Vincent et al. [19] reported that saponins extracted from TT had a beneficial effect on the control of poultry diseases. Feeding 1 TT powder/kg feed [20] or 0.06 and 0.12 g TT extracts/kg [21] had no significant effect on the growth performance of broiler chickens. In contrast, a 0.8 g TT powder/kg diet can improve the growth performance of broilers [22]. In another study, TT fruit powder was safe for the growth performance of mice up to a dose of 1 g/kg [2]. Sharma et al [23] found that the use of TT extract reduced functional renal disorders and cellular damage in rats. Liver function tests were increased as a result of cellular damage and degenerative changes with the increased permeability of the plasma membrane in the liver of broilers that received an overdose of TT [24]. Other studies reported that TT fruit powder can be used to treat liver regeneration and digestive problems and diarrhea in humans [25,26]. 

Although studies have shown some effects of TT, there is still much debate about possible mechanisms of action and therapeutic applications, as well as a significant lack of research data on this medicinal plant. To our knowledge, there is a limited number of studies in which the effects of adding TT powder as a whole plant to the diet on the performance and health status of broiler chickens has been examined. All previous findings are existing hypotheses from which this study was created. Therefore, the aim of the current study was to evaluate the use of TT powder on the performance, histological changes, and functional markers of the liver, kidney, and small intestine in broilers.

## 2. Materials and Methods

The use of birds, sampling and analyses were performed in accordance with the recommendations of the Scientific Research Ethics Committee of King Saud University, Riyadh, Saudi Arabia (approval number: KSU-SE-21-47).

### 2.1. Preparation of Tribulus Terrestris (TT) Powder

The TT was collected as a whole plant (flowers, seeds, leaves, and stems) in the valleys of the Republic of Yemen. The plants were classified in the herbarium of the College of Science-King Saud University (NO. 24519). Then, the plants were dried at a normal temperature (27 °C) until they reached a constant weight and were then ground into powder at the Research Department of Animal Production-Food and Agriculture Sciences at King Saud University.

### 2.2. Chemical Composition Profiles of Tribulus Terrestris (TT) Powder

The analysis of TT powder was performed in duplicate to determine the content of nutrients (dry matter, crude protein, ash, and total fiber) according to the methods of AOAC [27]. The fatty acids composition of TT powder was performed using the method of extraction with a mixture of chloroform and methanol, and then the total fat (g/100 g of TT) was calculated [28]. Fatty acid profiles were analyzed using a gas chromatograph mass spectrometer (GC-MS) [29]. Fatty acids were calculated based on the peak areas of the chromatogram and expressed as the percentage of fatty acid methyl esters. For the analysis of phenolic compounds, ethanol and methanol TT powder extracts were prepared according to the method of Al-Fatimi et al. [30]. Phenolic compounds were analyzed using high performance liquid chromatography (HPLC). Resorcinol, dinitrobenzene, chlorogenic acid, caffeic acid, vanillin, acetylsalicylic acid, salicylic acid, and quercetin were used as external standards. The amounts were expressed as ng/μL of extracted TT.

### 2.3. Birds and Experimental Design

One-day-old commercial Ross chickens (168 females in total) were used for this experiment. Birds were weighed individually and randomly divided equally into 3 treatments with 8 replicates per treatment (7 chicks per replicate) in an environmentally controlled battery with automatic electric heating. The initial temperature of 35 °C at one day of age was gradually reduced to 22 °C at 21 days of age and then maintained at this temperature and 55% humidity for the remainder of the 35-day period. Dietary treatments were supplemented with TT at three doses (C = 0.0, T1 = 0.75, and T2 = 1.5 g/kg feed). The basal diet for the chickens during the starter and finisher periods was formulated according to the recommendations from the commercial practice (Saudi Arabia) (see Table 1). Feed and water were offered ad libitum to the chickens during the experimental period. All chickens were vaccinated against NDV, IBV, and IBDV according to the manufacturer’s instructions (Fort Dodge Animal Health, Overland Park, KS, USA).

### 2.4. Live Performance Measurements 

All chicks were weighed individually to determine their body weight at days 1 and 35 and to calculate body weight gain per treatment. Feed intake was recorded during the experimental periods to calculate the feed conversion ratio. 

### 2.5. Measurement of Enzyme Activity

At 14 and 35 days of age, blood samples were collected from 16 birds in each dietary treatment in tubes without EDTA to measure liver function tests (alanine aminotransferase (ALT) and aspartate aminotransferase (AST)) and kidney function tests (uric acid and creatinine). Serum was separated via centrifugation at 3000× *g* for 20 min and then frozen at −80 °C until analysis. These functional tests were analyzed spectrophotometrically (RANDOX, London, UK) using reagent kits (Randox, London, UK) according to the manufacturer’s instructions. 

### 2.6. Morphology, Histological Changes 

At 14 and 35 days of age, the entire small intestine, liver, and kidney were randomly collected from 8 birds per treatment, weighed and calculated as a percentage based on live weight. The measurement of small intestine length was recorded and the relative length of intestinal segments was calculated based on the total small intestine length. The length (L) and weight (W) of the small intestine were used to calculate the W/L ratio [31]. 

The mid-portions of the ileum (1.5 cm) between the Meckel’s diverticulum and the cecal junction and the liver and kidneys (0.75 cm) of each bird were taken without stressing the tissue wall. Tissue samples were washed in phosphate-buffered saline (pH = 7.4) and fixed in 10% phosphate-buffered formalin for 72 h, then dehydrated in graded ethyl alcohol (50% to 100%) and paraffin-embedded into the action sections (Tissue-Tek, Sakura, Tokyo, Japan). Approximately 4 μm was cut using the Microtome System (Leica, RM 2245, Wetzlar, Germany). Sections were deparaffinized, hydrated and stained in hematoxylin and eosin (H&E) with Alcian blue stain to count the goblet cells (Leica, CV5030, Germany) using a protocol based on the modified method of Winsor and Sluys [32]. 

Tissue sections were examined under a light microscope (Nikon, Eclipse i80, Tokyo, Japan), and the required images were taken at various magnifications using a Nikon digital camera (OXM 1200C, Nikon, Tokyo, Japan). Measurements of villus length, width, depth, goblet cells, epithelial thickness, mucosa, submucosa, and lamina propria were based on at least ten intact villi from one section (bird) with a total of 80 measurements per treatment. The following equation based on villus length and width was used to calculate villus surface area (surface area = 2π × (width/2) × length) according to Al-Baadani et al. [33]. In addition, the ratio of villus length to crypt height was calculated [31]. Goblet cell density per 100 µm of villus area was determined according to the method of Qaid et al. [34]. Histopathological changes in the ileum, liver, and kidney in the different treatments were examined microscopically (100 to 400×) according to the method described by Belote et al. [35].

### 2.7. Statistical Analysis

The data obtained from the measurements of performance, enzymes, and morphology in the different treatments were statistically analyzed using the GLM procedures of SAS software [36] via one-way analysis ANOVA for a completely randomized design. Means were analyzed using Duncan’s multiple range test and treatments, with statistical significance being based on *p* < 0.05. All mean values for each parameter within the different treatments were reported as ± standard error of the means (SEM).

## 3. Results

The nutritional values and fatty acids composition of TT are presented in Table 2. The phytochemical compositions of TT included dry matter (92.84%), crude protein (15.52%), ash (17.19%), crude fiber (35.55%), and total fat (2.80%). As a result of the fatty acid profile, the saturated fatty acids and unsaturated fatty acid amounts of TT oil extract were found to be 19.38% and 80.62%, respectively. According to the measurement results, a total of nine various fatty acids for each saturated and unsaturated were determined in TT oil extract, including palmitic acid (C16:0; 11.44%), oleic acid (C18:1; 13.76%), and linoleic acid (C18:2; 64.32%).

The results of HPLC analysis (Table 3) showed that the phenolic content of TT is resorcinol, 1,2-Dinitrobenzene, chlorogenic acid, caffeic acid, vanillin, acetyl-salicylic acid, and salicylic acid, and the flavonoid content is quercetin. The major compound was acetylsalicylic acid at the rate of 160.99 or 524.86 ng/μL either in the ethanol or methanol extract, respectively. 

The results of the effect of TT on growth performance during the experimental period (1 to 35 days) are presented in Table 4. The present study showed that the intake of the highest level of TT at 1.5 g/kg diet (T2) resulted in lower (*p* ≤ 0.05) body weight gain and total feed intake compared to the basal diet (control), while the intake of TT at 0.75 g/kg diet (T1) had no negative effect. The feed conversion ratio was not affected by any of the treatments (*p* > 0.05).

The body weight and absolute and relative weights of the liver and kidney in broiler chickens fed TT powder are shown in Table 5. The body weight during slaughter data obtained showed that the chickens fed 0.75 g TT/kg diet (T1) displayed no effects (*p* > 0.05) compared to the control, while the chickens fed 1.5 g/kg had weights lower than those in the other treatments at 14 days of age (*p* < 0.05). At 35 days of age, the live body displayed no effects in all treatments (*p* > 0.05). The absolute and relative weights of the liver and kidney displayed no effects (*p* > 0.05) between all treatments during the starter and finisher phase (14 and 35 days). 

The effects of TT supplementation on liver and kidney function tests in broiler chickens are shown in Table 6. The measured liver function tests (ALT; alanine aminotransferase and AST; aspartate aminotransferase) and kidney function tests (uric acid and creatinine) showed no significant effect (*p* > 0.05) between the dietary treatments and the control group at 14 days of age. At 35 days of age, the chickens receiving the 0.75 g TT/kg diet (T1) had lower values (*p* < 0.05) on ALT and AST compared to those in the other dietary treatments. In contrast, there were no effects on uric acid and creatinine for all treatments (*p* > 0.05).

The results of histopathological changes in the livers of broiler chickens are shown in Figure 1, stained with (H&E) 100× to 400×. The livers of chickens fed a basal diet (control) showed the normal distribution of hepatocytes and dilation of the central vein. However, the examination of many chicken tissues revealed focal inflammatory infiltration by lymphocytes between the hepatocytes and around the central vein. The livers of chickens fed the 0.75 g TT powder/kg diet (T1) showed more normal hepatocytes compared with the control group, but dilation of the hepatic sinusoids located between the hepatic cords and narrowing of the central vein. In addition, the livers of T2 chickens (1.5 g TT powder/kg diet) showed normal hepatocytes but swollen hepatocytes in some areas, with lymphocyte infiltration around the central vein and the loss of lining endothelial cells.

Histopathological changes in the kidneys of broiler chickens are shown in Figure 2, stained with (H&E) 100× to 400×. Many renal tissues showed interstitial hemorrhage and congestion in renal blood vessels with the degeneration of the epithelial lining of renal tubules with the accumulation of inflammatory cells in chickens fed the basal diet (control). However, all histopathological changes in the control group improved in the tissues of the chickens when they received the 0.75 and 1.5 g TT powder/kg diet (T1 and T2, respectively).

The intestinal morphology, such as the weight and length of the small intestine, the ratio between the weight and length of the intestine, and the relative length of the small intestinal fragments in broilers fed diets enriched with TT powder, are shown in Table 7. At 14 days of age, the data revealed that the chickens fed a 0.75 g TT/kg diet (T1) had higher weights and lengths of small intestine (*p* < 0.05) than the control group. In addition, T1 and the control group had higher intestinal weight to intestinal length ratios and relative lengths of the duodenum (*p* < 0.05) compared to the chickens fed the 1.5 g/kg diet (T2). In contrast, the relative length of the ileum was lower (*p* < 0.05) in T1 and the control group. The small intestine weight and intestine weight to length ratio (W/L) were higher (*p* < 0.05) in chickens fed the 0.75 g TT/kg diet (T1) than in the control group, but there was no significant difference from T2 at 35 days of age.

Table 8 shows that the histomorphometric measurements of the ileum in broiler chickens were affected by the treatments (*p* < 0.001), while the number of goblet cells and epithelial thickness showed no differences between the treatments compared to the control (*p* > 0.05). The length and width of ileal villi and villus surface area were higher (*p* < 0.05) in chickens fed the 0.75 g TT/kg diet (T1) than T2 but were not significantly different between the T1 and control groups. In addition, the results showed that the ratio of villus length to crypt depth and the thickness of mucosa and lamina propria were higher (*p* < 0.05) in T1 chickens than in the other diets (T2 and control). In contrast, the crypt depth was lower in T1 compared to the other feed treatments, and the number of goblet cells per 100 µm villus area was also lower (*p* < 0.05) but not significantly different from the chickens fed the basal diet (control). 

The histopathological changes in the ileum in broiler chickens are shown in Figure 3. In the chickens fed the basic diet (control), the height and width of the villi were within the normal range, except for some cases where proliferative enterocytes with metaplasia of goblet cells were seen with disturbances in the villous spaces and the lumen contained sheets of epithelial cells, mucus, and denuded tips (Figure 3; control—100× and 300×). In birds fed a basal diet containing 0.75 g TT/kg feed (T1), normal heights and widths of the intestinal villi and a free lumen without any exudate, as well as a normal appearance of the submucosal and muscular sheaths were observed (Figure 3; T1—100× and 300×). In chicken intestines at T2 (1.5 g TT/kg feed), villi were shorter, and some villi fused due to the proliferation of enterocytes. Some villi showed the necrosis of enterocytes with numerous inflammatory cells. The lumen contained layers of epithelial cells that are mucinous (Figure 3; T2—100× and 300×).

## 4. Discussion

TT is a pharmaceutical herb that has long been used by people as a traditional medicine to treat many diseases, including urinary tract infections, to remove kidney urolithiasis, hypercholesterolemia, and liver health, and as a diuretic medication. These traditional uses would be due to the substances it contains with potential biological activity, including steroidal glycosides (saponins), protodeacon, glycosides, alkaloids, flavonoids, and tannins, as shown by previous studies [8,10,14,37]. The results of the current study show that the chemical composition of TT powder as a whole plant was rich in dry matter, crude protein, ash, and especially crude fiber (35.55%). Moreover, HPLC analysis showed that the major compound of TT was acetylsalicylic acid, in both ethanol and methanol extracts. The results of HPLC analysis of the chemical compounds in the extracts of TT in the current study agree with [1,2], who found that TT contains many pharmacologically active components belonging to flavonoids and phenols. However, the chemical composition of the extract varies depending on the extraction method and plant parts [38,39]. Several studies suggest that TT has preventive and therapeutic effects on many diseases, possibly due to the components it contains, such as phenolics and flavonoids [40,41]. Despite the diversity of saturated and unsaturated fatty acids in TT extract, but surprisingly, the majority of oils from TT seem to be composed of unsaturated fatty acids. Therefore, analysis revealed that palmitic acid (11.44%), oleic acid (13.76%), and linoleic acid (64.32%) were the major fatty acid components in TT extracts. These results are in agreement with those of Çömlekçioğlu and Çırak [42], who reported that the major fatty acids in TT plant are palmitic acid, oleic acid, and linoleic acid. These main fatty acids extracted from TT have many health benefits, anti-inflammatory properties, and promote metabolism [43,44]. Linoleic acid is an omega-6 polyunsaturated fatty acid and one of the essential fatty acids that should be consumed in the diet [45]. 

The results obtained show that the addition of TT powder (0.75 g/kg) to a basal diet did not affect overall growth (weight gain, feed intake and the feed conversion ratio) compared to chickens fed a basal diet from 1 to 35 days of age, while performance decreased when they received a high amount of TT powder (1.5 g/kg). Our results are in agreement with those of Ammar et al. [2], who showed that TT was safe for the growth performance of mice up to a dose of 1 g/kg. Another study by Şahin [22] reported that TT powder (0.80 g/kg feed) can be used to improve the growth performance of broiler chickens. In addition, the aqueous extract or powder of TT (5 and 10 mg/kg body weight) had positive effects on the growth performance of laying hens [11]. In contrast, dietary supplementation with TT powder (1 g/kg) or TT extracts (0.06 and 0.12 g/kg) did not affect the growth performance of broilers [3,21]. These results indicate that the over dosage of TT powder has a negative effect on growth performance in broiler chickens. This may be due to side effects of the high level of TT supplementation powder on the intestine, especially the ileum surface and measurements. Hayirli et al. [46] reported that TT powder at a dosage of 0.2 or 0.4 mg/kg had no positive effect on feed intake in broilers. The lower feed intake in T2 (1.5 g TT powder/kg diet) could be due to the high content of dietary fiber and phenolic compounds in TT or other factors. 

There were no effects on the absolute and relative weights of liver and kidney in broilers fed diets supplemented with TT powder (T1 and T2). Our results agree with the findings of Martino-Andrade et al. [47], who reported that the absolute and relative weights of liver and kidney were not different in all treated male rats compared to the control. On the other hand, Şahin and Duru [48] reported that dietary supplementation with TT extract (0.36 g/kg) decreased the weight of liver in broiler chickens. In liver function tests (ALT and AST) and kidney function tests (uric acid and creatinine), TT showed no effect at 14 days of age compared with the control group. The current results show that there was no difference in the function tests of the liver and kidney. At 35 days of age, chickens receiving T1 had lower levels of ALT and AST, while uric acid and creatinine showed no effect in all treatments. These results may be attributed to the concept that the diet supplemented with TT powder is not sufficient to induce changes in hepatocytes and cause renal dysfunction. Meanwhile, we found that examining histological changes in liver and kidney tissue of chickens that received TT powder (0.75 g/kg) showed normal morphological results, so it could be that TT powder at 0.75 g/kg has beneficial health effects in broiler chickens. The above results are consonant with those of Yazdi et al. [49], who reported that TT reduced histological damage and improved kidney function in rats. Furthermore, acetylsalicylic acid has no effect on human kidney function [50]. The results of this study are not in agreement with those of Anand et al. [19], who concluded that chickens fed a TT overdose had higher levels of ALT and AST, which were attributed to cellular damage, degenerative changes in the liver, and the increased permeability of the plasma membrane. In addition, acetylsalicylic acid contained in TT may play a beneficial role by reducing liver inflammation and oxidative stress [46]. However, lower levels of ALT and AST in chickens that received TT, especially in T1 (0.75 TT/kg feed may be attributed to a more normal liver histological structure compared with the control group. These were detected in the histological changes in the liver, although it seems that the changes in liver cells and structure did not affect the health of the birds so much, and the kidneys also showed improvement when the chickens received TT powder. Miranda et al. [51] concluded that the inclusion of TT (0.25, 0.50, and 0.75 g/kg diet) had safe effects on the liver and kidneys since no histopathological changes were observed in rats. 

According to our results on intestinal morphology, such as the weight and length of the small intestine and the ratio of weight to length of the intestine, the absorption area was higher when the diet was supplemented with TT, especially at T1 (0.75 g/kg). Moreover, histomorphometric measurements of the ileum in broiler chickens were improved when they received TT powder, whereas ileal villus length and width, surface area, villus length to crypt height ratio, mucosa, and lamina propria thickness were lower in chickens receiving 1.5 g TT/kg feed (T2). Crypt height is an indicator of the multiplication and absorption capacity of the intestine [52]. Previous reports have shown that as villus height increases, both the digestive and absorptive functions of the intestine increase, resulting in an increased absorptive surface area, the increased expression of brush border enzymes, nutrient transport systems, and body weight [53]. However, many substances can affect the development of the inner villi, making enzymatic activity and enterocyte structure two essential features of the physiology of the intestinal mucosa [54]. The intake of the 1.5 g TT/kg diet resulted in a decrease in the length, width, and surface area of villi, which may be due to the suppression of beneficial bacteria in the intestine, thus reflected in the decreased growth performance. Mucus production is related to the density of goblet cells and is an important component of the intestinal barrier [55]. However, the results obtained showed no differences in goblet cell density between all treatments. Changes in the histopathology of the ileum of broiler chickens fed the basal diet and 1.5 g TT/kg diet (T2) were observed; shorter villi were observed with the fusion of some villi, which could be due to proliferation of enterocytes with enterocyte necrosis, and the lumen contains sheets of epithelial cells that are slimy. The use of TT decreased the crypt depth and increased the ratio between villus height and crypt depth in the ileum of the broiler. The higher this ratio, the greater the ability to digest and absorb nutrients [56]. These results related to changes in gut histology, which may have led to decreased growth performance in chickens fed the basal diet containing 1.5 g TT/kg feed (T2). Supplementing the diet with a high dose of medicinal plants may have negative effects on some beneficial microbial populations [57]. In addition, birds fed a 1.5 g TT/kg diet may suffer from the negative effects on the intestinal epithelium, resulting in mucosal damage. Therefore, the most common standards for assessing abdominal mucosal injury and health in poultry are villus length, crypt depth, and villus/crypt ratio [58,59]. The villus crypt is considered a villus factory, and deeper crypts indicate rapid tissue turnover, allowing villi to be renewed as needed in response to normal shedding [60]. A reduction in villus height and deeper crypts may result in poor nutrient uptake, increased secretion into the internal tract, and decreased performance [61]. In contrast, Awad et al. [62] reported that an increase in villus length and the villus-to-crypt ratio correlated with increased epithelial cell turnover, and longer villi were associated with activated cell mitosis. 

## 5. Conclusions

The current results conclude that the use of TT powder as a whole plant contains nutritional compositions and saturated fatty acids as well as pharmacologically active components. Therefore, the addition of TT in a 0.75 g/kg diet had no negative effects on growth performance and liver and kidney morphology and improved intestinal morphology and ileum histomorphometry. Furthermore, adding TT powder (0.75 g/kg) to the diet improved the histological changes in the liver, kidneys, and ileum. The dosage of 0.75 g/kg diet used in the current study is likely to be safe and beneficial for the health of broiler chickens. However, further studies are needed to discover the proper dosages and mechanism of pharmacological effect of TT powder as a whole plant on the physiological and health characteristics of broiler chickens. 

## Figures and Tables

**Figure 1 animals-12-01930-f001:**
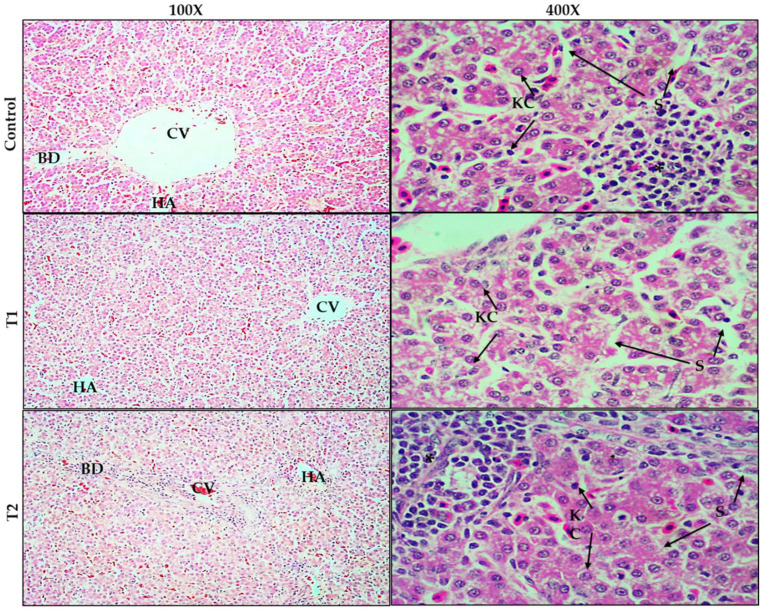
Histological changes in the liver of broiler chickens stained with (H&E) 100× to 400×. Control, birds fed basal diet; T1, basal diet containing 0.75 g *Tribulus Terrestris*/kg feed; T2, basal diet containing 1.5 g *Tribulus Terrestris*/kg feed. CV = central vein; S = sinusoids; KC = Kupffer cells; HA = hepatic artery; BD = bile duct. Chicken liver (control): shows normal morphology of hepatocytes with focal inflammatory infiltration by leukocytes (lymphocytes) (star) and dilatation of the CV. Chicken liver (T1): hepatocytes are more normal but sinusoidal spaces are dilated (S) and CV is narrower than control group. The chicken liver (T2): hepatocytes normal, but swollen hepatocytes in some areas with lymphocyte infiltration around the central vein (star) and loss of lining endothelial cells.

**Figure 2 animals-12-01930-f002:**
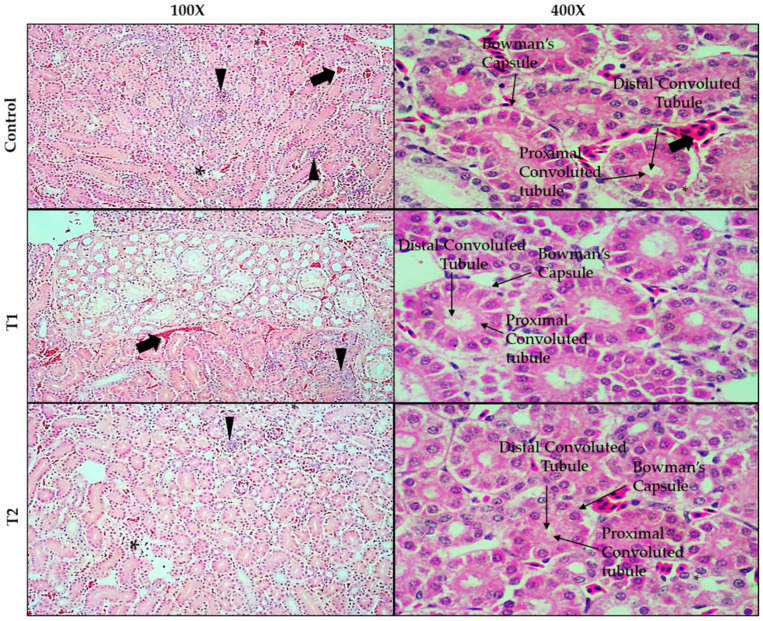
Histological changes in the kidney of broiler chickens stained with (H&E) 100× to 400×. Control, birds receiving basal diet; T1, basal diet containing 0.75 g *Tribulus Terrestris*/kg feed; T2, basal diet containing 1.5 g *Tribulus Terrestris*/kg feed. The kidney of the chickens (control) showed interstitial hemorrhage and congestion in the renal blood vessels (striped arrow) with degeneration of the epithelial lining of the renal tubules (star) with accumulation of inflammatory cells (arrowhead). In contrast, all histopathological changes improved in chickens fed diets T1 to T2 (0.75 or 1.5 g TT powder per kg diet).

**Figure 3 animals-12-01930-f003:**
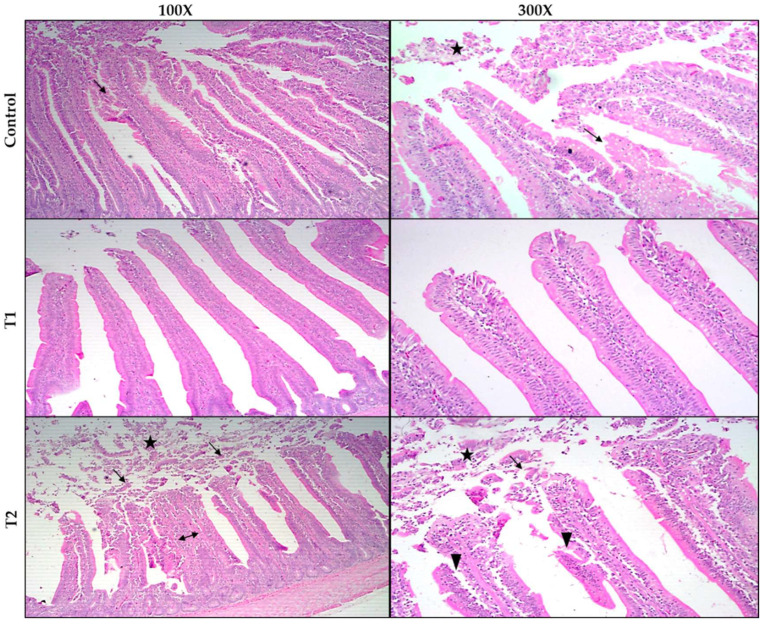
Histological changes in the ileum of broiler chickens stained with (H&E) 100× to 300×. Control, birds fed basal diet; T1, basal diet containing 0.75 g *Tribulus Terrestris*/kg feed; T2, basal diet containing 1.5 g *Tribulus Terrestris*/kg feed. Chicken ileum (control): showing normal height and width of intestinal villi with proliferative enterocytes (arrows). The lumen contains layers of epithelial cells (star). The chicken ileum (T1): normal height and width of the intestinal villi and the lumen free of any exudate and greater improvement than observed in the other treatment groups. Chicken ileum (T2): short villi with fusion of some villi (double arrow). Some villi show necrosis of enterocytes (arrowhead) and contain sheets of epithelial cells and mucus in the lumen (star) as well as proliferative enterocytes (arrows).

**Table 1 animals-12-01930-t001:** Feed ingredients and calculated content of the basal diet.

Ingredients	Basal Diet (%)	
	Starter (Days 1–14)	Finisher (Days 15–35)
Corn	54.89	60.13
Soybean meal	38.77	31.92
Palm oil	1.96	4.21
Dicalcium phosphate	1.74	1.38
Limestone	1.19	1.05
Salt	0.40	0.40
Min. Vit. Premix, 0.5% ^a^	0.50	0.50
DL-Methionine	0.31	0.25
L-Lysine-HCL	0.13	0.08
L-Threonine	0.07	0.02
Choline CL-70%	0.05	0.05
Total	100	100
Calculated composition		
Metabolizable energy, Kcal/kg	3000	3200
Crude protein	23.0	20.0
Crude fiber	2.21	2.10
Available P	0.48	0.41
Lysine	1.28	1.06
Methionine	0.64	0.55
Methionine + cysteine	0.95	0.83
Threonine	0.86	0.71
Tryptophan	0.27	0.23
Arginine	1.50	1.29
Valine	1.13	0.99

^a^ Containing by kg of diets: 2,400,000 IU of vitamin A; 1,000,000 IU of vitamin D; 16,000 IU of vitamin E; 800 mg of vitamin K; 600 mg of vitamin B1; 1600 mg of vitamin B2; 1000 mg of vitamin B6; 6 mg of vitamin B12; 8000 mg of niacin; 400 mg of folic acid; 3000 mg of pantothenic acid; 40 mg of biotin; 80 mg of cobalt; 2000 mg of copper; 400 mg of iodine; 1200 mg of iron; 18,000 mg of manganese; 60 mg of selenium and 14,000 mg of zinc.

**Table 2 animals-12-01930-t002:** Nutritional values and fatty acids composition of *Tribulus Terrestris* (TT).

Item	%
Chemical composition ^1^	
Dry matter	92.84
Crude protein	15.52
Ash	17.19
Crude fiber	35.55
Total fat	2.80
Fatty acids of total fat ^2^	
C6:0	0.15
C8:0	0.34
C12:0	0.08
C14:0	0.21
C16:0	11.44
C17:0	0.17
C18:0	6.86
C22:0	0.03
C20:0	0.10
Saturated fatty acids	19.38
C16:1	0.11
C18:1	13.76
C18:2	64.32
C18:3 (gamma)	0.54
C18:3 (α-linoliec)	1.42
C20:2	0.09
C20:3	0.03
C20:4	0.16
C22:5	0.19
Unsaturated fatty acids	80.62

^1^ The chemical composition analysis was performed in duplicate on a dry matter basis; ^2^ The fatty acids composition was analyzed via GC-MS in duplicate.

**Table 3 animals-12-01930-t003:** Phenolic and flavonoid compounds of *Tribulus Terrestris* (TT). The phenolic compounds were analyzed via HPLC in duplicate.

Compounds	Ethanol Extract			Methanol Extract		
	Area (mAU*s)	Ret Time (min)	Amount (ng/μL)	Area (mAU*s)	Ret Time (min)	Amount (ng/μL)
Resorcinol	117.7	7.33	10.24	341.40	7.36	29.58
1,2-Dinitrobenzene	492.3	9.34	31.16	98.90	8.99	6.30
Chlorogenic acid	388.4	12.02	23.23	126.90	11.77	7.45
Caffeic acid	539.6	14.13	12.10	1008.50	14.19	22.37
Vanillin acid	1065.4	16.30	19.65	1773.10	16.37	32.53
Acetyl salicylic acid	278.3	22.17	160.99	915.80	22.27	524.86
Salicylic acid	470.5	24.95	32.70	487.30	24.77	36.17
Quercetin	92.6	34.80	33.37	235.74	34.89	91.98

**Table 4 animals-12-01930-t004:** Effect of *Tribulus Terrestris* (TT) supplementation on growth performance in broiler chickens from 1 to 35 days of age.

Variables	Treatments ^1^			SEM ^2^	*p*-Value
	Control	T1	T2		
Bodyweight gain, g	1714.02 ^a^	1716.76 ^a^	1630.41 ^b^	14.12	0.0003
Total feed intake, g	2435.97 ^a^	2350.63 ^a,b^	2323.31 ^b^	32.58	0.050
Feed conversion ratio, g/g	1.42	1.36	1.42	0.018	0.072

^a,b^ Means values within a row for each variable with clarification of the significant difference in the form of superscripts (*p* < 0.05). ^1^ Treatments: control, birds fed basal diet; T1, basal diet with 0.75 g *Tribulus Terrestris*/kg diet; T2, basal diet with 1.5 g *Tribulus Terrestris*/kg diet. ^2^ SEM = standard error of means for treatment effect.

**Table 5 animals-12-01930-t005:** Effect of *Tribulus Terrestris* (TT) supplementation on absolute and relative weight of liver and kidney in broiler chickens.

Variables	Treatments ^1^			SEM ^2^	*p*-Value
	Control	T1	T2		
At14 days of age					
Body weight (g)	438.75^a^	429.88 ^a^	369.38 ^b^	15.47	0.009
Liver (g)	12.74	10.81	9.86	0.89	0.090
Liver (%)	2.94	2.50	2.67	0.21	0.363
Kidney (g)	2.26	2.90	2.60	0.25	0.238
Kidney (%)	0.528	0.672	0.710	0.06	0.149
At 35 days of age					
Body weight (g)	1672.63	1728.00	1693.75	21.66	0.238
Liver (g)	35.68	38.33	35.72	1.25	0.250
Liver (%)	2.13	2.21	2.11	0.09	0.672
Kidney (g)	2.32	2.40	1.99	0.19	0.319
Kidney (%)	0.14	0.13	0.11	0.01	0.391

^a,b^ Means values within a row for each variable with clarification of the significant difference in the form of superscripts (*p* < 0.05). ^1^ Treatments: control, birds fed basal diet; T1, basal diet with 0.75 g *Tribulus Terrestris*/kg diet; T2, basal diet with 1.5 g *Tribulus Terrestris*/kg diet. ^2^ SEM = standard error of means for treatment effect.

**Table 6 animals-12-01930-t006:** Effect of *Tribulus Terrestris* (TT) supplementation on function tests of liver and kidney in broiler chickens.

Variables	Treatments ^1^			SEM ^2^	*p*-Value
	Control	T1	T2		
At 14 days of age					
AST U/L	171.5	172.5	173.8	3.13	0.874
ALT U/L	14.6	13.2	12.3	2.09	0.732
Uric acid mg/dl	2.12	2.43	2.46	0.18	0.362
Creatinine mg/dl	0.444	0.452	0.363	0.04	0.387
At 35 days of age					
AST U/L	245.2 ^a^	208.7 ^c^	227.9 ^b^	3.39	<0.0001
ALT U/L	17.5 ^a^	9.3 ^c^	11.1 ^b^	2.29	0.050
Uric acid mg/dl	1.82	1.63	1.48	0.22	0.584
Creatinine mg/dl	0.365	0.414	0.359	0.04	0.647

^a–c^ Means values within a row for each variable with clarification of the significant difference in the form of superscripts (*p* < 0.05). ^1^ Treatments: control, birds fed basal diet; T1, basal diet with 0.75 g *Tribulus Terrestris*/kg diet; T2, basal diet with 1.5 g *Tribulus Terrestris*/kg diet. ^2^ SEM = standard error of means for treatment effect.

**Table 7 animals-12-01930-t007:** Effect of *Tribulus Terrestris* (TT) supplementation on the intestinal morphology in broiler chickens.

Variables	Treatments ^1^			SEM ^2^	*p*-Value
	Control	T1	T2		
At14 days of age					
Small intestine weight (g)	19.33 ^b^	23.61 ^a^	18.09 ^b^	1.08	0.004
Small intestine length (cm)	118.75 ^b^	131.56 ^a^	130.04 ^a^	2.26	0.001
Small intestine length (%)					
Duodenum	17.93 ^a^	16.66 ^a,b^	16.13 ^b^	0.50	0.050
Jejunum	44.00	44.19	41.90	0.93	0.183
Ileum	38.06 ^b^	39.14 ^b^	41.96 ^a^	0.92	0.019
W/L ratio ^3^ (g/cm)	0.163 ^a^	0.177 ^a^	0.138 ^b^	0.007	0.007
At 35 days of age					
Small intestine weight (g)	55.91 ^b^	63.31 ^a^	60.26 ^a,b^	1.79	0.027
Small intestine length (cm)	177.38	176.21	180.53	3.53	0.676
Small intestine length (%)					
Duodenum	16.25	16.91	15.97	0.42	0.303
Jejunum	42.38	42.28	43.32	0.43	0.197
Ileum	41.37	40.80	40.69	0.53	0.638
W/L ratio ^3^ (g/cm)	0.313 ^b^	0.360 ^a^	0.333 ^a,b^	0.009	0.009

^a,b^ Means values within a row for each variable with clarification of the significant difference in the form of superscripts (*p* < 0.05). ^1^ Treatments: control, birds fed basal diet; T1, basal diet with 0.75 g *Tribulus Terrestris*/kg diet; T2, basal diet with 1.5 g *Tribulus Terrestris*/kg diet. ^2^ SEM = standard error of means for treatment effect. ^3^ W/L ratio: intestine weight to length ratio.

**Table 8 animals-12-01930-t008:** Effect of *Tribulus Terrestris* (TT) supplementation on histomorphometric measurements of ileum in broiler chickens.

Variables	Treatments ^1^			SEM ^2^	*p*-Value
	Control	T1	T2		
Length (μm)	419.17 ^a^	403.55 ^a^	307.47 ^b^	15.92	<0.0001
Width (μm)	49.76 ^a,b^	52.43 ^a^	44.87 ^b^	1.89	0.007
Villus surface area (mm^2^)	0.065 ^a^	0.066 ^a^	0.042 ^b^	0.002	<0.0001
Crypt-depth (μm)	26.96 ^a^	19.99 ^b^	26.12 ^a^	0.91	<0.0001
Villus length/crypt depth	15.67 ^b^	21.01 ^a^	11.85 ^c^	0.93	<0.0001
Goblet cells (NO.)	87.58	86.25	80.35	3.35	0.182
Goblet cells/100 µm Villi area	10.55 ^b^	10.73 ^b^	13.87 ^a^	0.87	0.004
Epithelial thickness (μm)	5.76	5.34	6.55	0.44	0.084
Mucosa (μm)	28.84 ^b^	33.61 ^a^	23.90 ^c^	0.76	<0.0001
Sub-mucosa (μm)	13.48 ^a,b^	15.25 ^a^	12.25 ^b^	0.67	0.003
Lamina propria thickness (μm)	19.33 ^b^	23.52 ^a^	14.78 ^c^	1.26	<.0001

^a–c^ Means values within a row for each variable with clarification of the significant difference in the form of superscripts (*p* < 0.05). ^1^ Treatments: control, birds fed basal diet; T1, basal diet with 0.75 g *Tribulus Terrestris*/kg diet; T2, basal diet with 1.5 g *Tribulus Terrestris*/kg diet. ^2^ SEM = standard error of means for treatment effect.

## Data Availability

All data sets collected and analyzed during the current study are available from the corresponding author on fair request.

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
