# Peer review of "Growth Performance, Histological Changes and Functional Tests of Broiler Chickens Fed Diets Supplemented with Tribulus Terrestris Powder"

_animals, 2022, doi:10.3390/ani12151930_

Round 1
Reviewer 1 Report
Author added all suggestions so it can accept
Reviewer 2 Report
The manuscript is now satisfactory
Reviewer 3 Report
All reviewer comments addressed carefully by authors in the revised paper. The revised paper is sound and is well structured, and follows a logical sequence.
So, based on my opinion the revised manuscript merits acceptance.
This manuscript is a resubmission of an earlier submission. The following is a list of the peer review reports and author responses from that submission.
Round 1
Reviewer 1 Report
The study investigated growth performance, histological changes and functional tests of broiler chickens fed diets supplemented with Tribulus Terrestris powder. I could not catch the aim of the investigation; what the authors explore the efficacy of TT and suggest from the results. They evaluated TT as an alternative to antibiotics in the feed; however, there is no data on intestinal microbiota and its-related parameters. A larger number of groups is needed to find the optimal dosage (also for overdose), and more positive data is also needed. There is little positive effect of TT supplementation, and how the additive is used even though decreased ALT and AST activities? From these, the study needs to be reformulated.
Author Response
Response to Reviewer Comments:    
**Reviewer #1
Q1: I could not catch the aim of the investigation; what the authors explore the efficacy of TT and suggest from the results.
Authors’ Response: Thank you for pointing out that. We revised as required with modified.
Q2: They evaluated TT as an alternative to antibiotics in the feed; however, there is no data on intestinal microbiota and its-related parameters.
Authors response: Thank you for your comment. We revised as required. But in fact, it may be an alternative to antibiotics if studies continue and we are in the process of doing so in the future.
Q3: A larger number of groups is needed to find the optimal dosage (also for overdose), and more positive data is also needed.
Authors response: Thank you for your feedback. In fact, we are in the process of conducting a future study to determine the ideal dose, since this study is preliminary and i think it is the only one that investigates the effect of TT on the histological changes and health status of broiler chickens, as a result of current study showed the threshold dose that affected on intestinal tissue and we planned to future work to find the optimal dose by a larger number of groups.
Q4: There is little positive effect of TT supplementation, and how the additive is used even though decreased ALT and AST activities?
Authors response: Thank you for pointing out that. Decreased ALT and AST activities in chickens that received TT, especially in T1 (0.75 TT/kg feed) within the normal range and does not represent a pathological condition when compared to control, where birds of the control group may be exposed to various factors while not being fed on a diet without any supplement. This was confirmed by the histological changes of the liver, which were more normal liver histological structures compared with the control group. We mentioned that in the discussion.
Q5: From these, the study needs to be reformulated.
Authors’ Response: Thank you for pointing out that. We revised as required in all manuscript
The manuscript has been completely revised, with some language changes made and improved from our point of view, on the other hand, I appreciate your efforts in your valuable comments and question, which gave me the opportunity to improve the throughout manuscript.
Please if there are any opinions, guide us to correct it.
satisfactory to you
Thanks so much for your efforts. Your feedbacks are very valuable and will improve my research skills and biological insight on my future studies.
--
Abdulmohsen H. Alqhtani
Reviewer 2 Report
In this manuscript, the researchers tried to explain about the Growth Performance, Histological Changes and Functional Tests of Broiler Chickens Fed Diets Supplemented with Tribulus Terrestris Powder. It is interesting work and can be accepted after revision.
- The grammar errors should be checked in the whole manuscript.
- In abstract, the first four lines should be summarized.
- In introduction, the main objective has been repeated so it should be refined.
- Some recent and relevant articles may be added as thousands of articles have been published on this topic.
- Conclusion should be refined as it is not properly written as per results.
Author Response
Response to Reviewer Comments:    
**Reviewer #2
In this manuscript, the researchers tried to explain about the Growth Performance, Histological Changes and Functional Tests of Broiler Chickens Fed Diets Supplemented with Tribulus Terrestris Powder. It is interesting work and can be accepted after revision.
Q1: The grammar errors should be checked in the whole manuscript.
Authors’ Response: Thanks for your feedback. We have improved grammar errors in all manuscript.
Q2: In abstract, the first four lines should be summarized.
Authors’ Response: Done as requested.
Q3: In introduction, the main objective has been repeated so it should be refined.
Authors’ Response: Thank you for your notes. Done as requested. Lines 78-80.
Q4: Some recent and relevant articles may be added as thousands of articles have been published on this topic.
Authors’ Response: Done as requested.
Q5: Conclusion should be refined as it is not properly written as per results.
Authors response: We revised as required with the Conclusion rewritten as per the results. Lines 428-437.
The manuscript has been completely revised, with some language changes made and improved from our point of view, on the other hand, I appreciate your efforts in your valuable comments and question, which gave me the opportunity to improve the throughout manuscript.
Please if there are any opinions, guide us to correct it.
satisfactory to you
Thanks so much for your efforts. Your feedbacks are very valuable and will improve my research skills and biological insight on my future studies.
--
Abdulmohsen H. Alqhtani

Reviewer 3 Report
The simple summary suggests (line 12) that TT is being used as an alternative to antibiotics but there is no diet/treatment where antibiotics are used, so how can you make any conclusions regarding this purported function? There is no mention of antibiotics in the rest of the manuscript. I cannot see any literature to suggest that TT has antimicrobial benefits, so the whole basis for this study appears to be unsubstantiated.
I have a lot of difficulty accepting many of the statements made in the Introduction and find the references cited to support such statements are not about those topics. For example: line 39 suggests that phytobiotic compounds are the same as plants when they are not. line 53 cites reference 8 as support for the use of TT as an effective medicine for a wide range of human disease conditions, but reference 8 is a study on the effect of TT on semen quality in roosters. You have to use primary literature as supporting evidence for such profound statements. Similarly, line 57 cites reference 11 as providing evidence that TT improves renal function in rats, it doesn't. Reference 11 is a review and it clearly concludes that the data supporting the efficacy of TT as a health supplement are lacking and the purported effects are unproven. It does say that this may be due to differences in processing practices and other non-standardised issues of study design, but it is clearly saying the claims are not proven.
Line 65 and elsewhere. There are statements that liver function tests are higher in the TT groups but it is not clear whether higher levels are a positive or negative for the health of the animal.
Line 69 suggests "there is no overdose study of TT in broilers" but on line 65 it said that there was such a study. Which statement is correct?
The sentence from line 69-71 does not make sense. I have no idea what you are trying to tell me.
Line 72. TT is purported to be a natural alternative, but an alternative to what? If it's an alternative then surely one of the treatment groups should contain the thing that it is an alternative to? But this is not the case.
Line 82 - is purified just the drying of the plant? If so there's no purification.
I have a real problem with the data in Tables 4 and 5. In one Table it shows total bodyweight gain, in the other it shows the live weight of the birds at day 35 (the end of the study). How could the final live weight of the Control birds be less than the total bodyweight gain?
The conclusion of this study is that TT at 0.75g/kg of feed does no harm to the birds, but the more important question should be 'is there any benefit in adding 0.75g/kg TT to the broiler diet'? I don't believe there is evidence to say that there is a benefit, so if there's no benefit why would you do it? If the authors believe it has a benefit then they must more clearly state what it is.
Author Response
Response to Reviewer Comments:    
**Reviewer #3
Q1: The simple summary suggests (line 12) that TT is being used as an alternative to antibiotics but there is no diet/treatment where antibiotics are used, so how can you make any conclusions regarding this purported function? There is no mention of antibiotics in the rest of the manuscript. I cannot see any literature to suggest that TT has antimicrobial benefits, so the whole basis for this study appears to be unsubstantiated.
Authors response: Thanks so much for your notification valuable. We have corrected as per the study. Line 13.
Q2: Line 39. suggests that phytobiotic compounds are the same as plants when they are not.
Authors response: Thanks so much for your notification. We have corrected this in line 42.
Q3: Line 53. cites reference 8 as support for the use of TT as an effective medicine for a wide range of human disease conditions, but reference 8 is a study on the effect of TT on semen quality in roosters.
Authors response: Thank you for your comments. Reference 8 reported that TT also has an effect on hypercholesterolemia, as well as it has been added references and conclusions about the use of TT as an effective medicine. Line 57.
Q4: Line 57. cites reference 11 as providing evidence that TT improves renal function in rats, it doesn't.
Authors’ Response: Thank you for your notes. We have corrected this reference. Line 61.
Q5: Line 65. and elsewhere. There are statements that liver function tests are higher in the TT groups but it is not clear whether higher levels are a positive or negative for the health of the animal.
Authors response: Thank you for pointing out that. In conditions of different levels of damage in hepatocytes, ALT from cytoplasm and AST from mitochondria leak out thus higher levels of ALT and AST are negative for the health of the animal. The AST and ALT in current study lied on the normal range with no evidence effect of the liver function. However, more normal liver histological structures in the dietary supplemented with TT compared with the control group. We mentioned that in the discussion.
Q6: Line 69. suggests "there is no overdose study of TT in broilers" but on line 65 it said that there was such a study. Which statement is correct?
Authors response: Thank you for your notes. it has been modified. Line 73.
Q7: Line 69-71. does not make sense. I have no idea what you are trying to tell me.
Authors’ Response: Done as requested. Line 73-78.
Q8: Line 72. TT is purported to be a natural alternative, but an alternative to what? If it's an alternative then surely one of the treatment groups should contain the thing that it is an alternative to? But this is not the case.
Authors response: Thank you for your notes. it has been modified. Line 78.
Q9: Line 82. is purified just the drying of the plant? If so there's no purification.
Authors’ Response: There was an error. It has been modified. Line 88.
Q10: I have a real problem with the data in Tables 4 and 5. In one Table it shows total bodyweight gain, in the other it shows the live weight of the birds at day 35 (the end of the study). How could the final live weight of the Control birds be less than the total bodyweight gain?
Authors’ Response: Thanks for your feedback. Table 5 is the bodyweight of the birds that were randomly selected without bias or considering the average cage (8 replicates per treatment; 7 chicks per replicate) for slaughter so that one bird as an experimental unit for histological measurements and the relative rate of the target organs.
The bird that was selected is among the 7 birds for the repeater (cage), which represents an experimental unit for performance parameters.
In other words, the body weight in table 5 is not included in the calculation of the weight gain in the performance table (4) because the number of birds varies to calculate the average.
However, the live weight during the slaughter may be less than the total bodyweight gain.
Q11: The conclusion of this study is that TT at 0.75g/kg of feed does no harm to the birds, but the more important question should be 'is there any benefit in adding 0.75g/kg TT to the broiler diet'? I don't believe there is evidence to say that there is a benefit, so if there's no benefit why would you do it? If the authors believe it has a benefit then they must more clearly state what it is.
Authors’ Response: Thank you for this opinion, we have corrected this concept by defining the importance and conclusion of this study more clearly.
Please if there is another opinion, guide us to correct it.
The manuscript has been completely revised, with some language changes made and improved from our point of view, on the other hand, I appreciate your efforts in your valuable comments and question, which gave me the opportunity to improve the throughout manuscript.
Please if there are any opinions, guide us to correct it.
satisfactory to you
Thanks so much for your efforts. Your feedbacks are very valuable and will improve my research skills and biological insight on my future studies.
--
Abdulmohsen H. Alqhtani

Reviewer 4 Report
1- English should improve by a native person. The paper suffers from a poor English structure throughout and cannot be published or reviewed properly in the current format. The manuscript requires a thorough proofread by a native person whose first language is English. The instances of the problem are numerous and this reviewer cannot individually mention them. It is the responsibility of the author(s) to present their work in an acceptable format. Unless the paper is in a reasonable format, it should not have been submitted.
2- The novelty of the study needs to be highlighted compare to other similar studies or consider to explicitly mention what is gap knowledge and/or what was lacking in the indicated studies.
3- Discussion is weak. The discussion needs enhancement with real explanations not only agreements and disagreements. Authors should improve it by the demonstration of biochemical/physiological causes of obtained results. Instead of just justifying results, results should be interpreted, explained to appropriately elaborate inferences. discussion seems to be poor, didn't give good explanations of the results obtained. I think that it must be really improved. Where possible please discuss potential mechanisms behind your observations. You should also expand the links with prior publications in the area, but try to be careful to not over-reach. For the latter, you should highlight potential areas of future study.
4- The scientific background of the topic is poor. In "Introduction" and "Discussion", the authors should cite recent references between 2021-2022 from JCR journals (with impact factor) about recent achievements on the subject. For example, authors should cite to:
Nikbakht, S.A., Mohammadabadi, T. and Mirzadeh, K. (2021). The Effect of Feeding Tribulus terrestris Plant Powder on Growth Performance, Digestibility, Rumen and Blood Parameters of Iranian Arabic Lambs. Iranian J. Appl. Anim. Sci., 11(4), 781-788.
Masoudi, R., H. Javaheri Barfourooshi, S. A. Hosseini, F. Zarei, and Z. Abdollahi. "Effect of Tribulus terrestris, Ceratonia silique and Zingiber officinale on reproductive potential of Ross broiler breeder roosters." Veterinary Researches & Biological Products 34, no. 4 (2021): 177-186.5- A detailed "Conclusion" should be provided to state the final result that the authors have reached. Please note you only need to place your conclusion and not keep putting results, because these have already been presented in the manuscript.
6- Author(s) should re-format the references based on journal format. See the instructions for authors.
Author Response
Response to Reviewer Comments:    
**Reviewer #4
Q1: English should improve by a native person. The paper suffers from a poor English structure throughout and cannot be published or reviewed properly in the current format. The manuscript requires a thorough proofread by a native person whose first language is English.
Authors’ Response: Thanks for your feedback. We have improved the English and grammar errors in all manuscript.
Q2: The novelty of the study needs to be highlighted compare to other similar studies or consider to explicitly mention what is gap knowledge and/or what was lacking in the indicated studies.
Authors’ Response: Thank you for this constructive opinion. Done as requested in lines 71-76 and 436.
Q3: Discussion is weak. The discussion needs enhancement with real explanations not only agreements and disagreements. Authors should improve it by the demonstration of biochemical/physiological causes of obtained results. Instead of just justifying results, results should be interpreted, explained to appropriately elaborate inferences. discussion seems to be poor, didn't give good explanations of the results obtained. I think that it must be really improved. Where possible please discuss potential mechanisms behind your observations. You should also expand the links with prior publications in the area, but try to be careful to not over-reach. For the latter, you should highlight potential areas of future study.
Authors’ Response: Thank you for your notes. We re-formatted discussion as well as enhancement and explanations as much as possible. Also, highlight potential areas of future study in line 471. Please if there is another opinion, guide us to correct it.
Q4: The scientific background of the topic is poor. In "Introduction" and "Discussion", the authors should cite recent references between 2021-2022 from JCR journals (with impact factor) about recent achievements on the subject. For example, authors should cite to:
Nikbakht, S.A., Mohammadabadi, T. and Mirzadeh, K. (2021). The Effect of Feeding Tribulus terrestris Plant Powder on Growth Performance, Digestibility, Rumen and Blood Parameters of Iranian Arabic Lambs. Iranian J. Appl. Anim. Sci., 11(4), 781-788.
Masoudi, R., H. Javaheri Barfourooshi, S. A. Hosseini, F. Zarei, and Z. Abdollahi. "Effect of Tribulus terrestris, Ceratonia silique and Zingiber officinale on reproductive potential of Ross broiler breeder roosters." Veterinary Researches & Biological Products 34, no. 4 (2021): 177-186.
Authors response: Thank you for your notes valuable. it has been modified Introduction and Discussion with cite recent references (referred above).
Please if there is another notes, guide us to correct it.
Q5: A detailed "Conclusion" should be provided to state the final result that the authors have reached. Please note you only need to place your conclusion and not keep putting results, because these have already been presented in the manuscript.
Authors’ Response: Done as requested. Lines 433-442.
Q6: Author(s) should re-format the references based on journal format. See the instructions for authors.
Authors response: We re-formatted all references based on journal format.
The manuscript has been completely revised, with some language changes made and improved from our point of view, on the other hand, I appreciate your efforts in your valuable comments and question, which gave me the opportunity to improve the throughout manuscript.
Please if there are any opinions, guide us to correct it.
satisfactory to you
Thanks so much for your efforts. Your feedbacks are very valuable and will improve my research skills and biological insight on my future studies.
--
Abdulmohsen H. Alqhtani

Round 2
Reviewer 3 Report
In several lines that have been revised the T1 dose is shown as 0.7g/kg when it should be 0.75g/kg.
Also it should be possible to show a proper mu symbol instead of u in microliter.